# Can the Day of the Week and the Time of Birth Predict the Mode of Delivery According to Robson Classification?

**DOI:** 10.3390/healthcare11152158

**Published:** 2023-07-28

**Authors:** Paraskevi Giaxi, Kleanthi Gourounti, Victoria Vivilaki, Dimitra Metallinou, Panagiotis Zdanis, Antonis Galanos, Aikaterini Lykeridou

**Affiliations:** 1Department of Midwifery, University of West Attica, Agioy Spyridonos 28, 12243 Egaleo, Greece; kgourounti@uniwa.gr (K.G.); vvivilaki@uniwa.gr (V.V.); dmetallinou@uniwa.gr (D.M.); midw18681087@uniwa.gr (P.Z.); klyker@uniwa.gr (A.L.); 2Laboratory for Research of the Musculoskeletal System, School of Medicine, National and Kapodistrian University of Athens, 14561 Athens, Greece; galanostat@yahoo.gr

**Keywords:** clinical audit, mode of birth, non-clinical factors, Greece, Robson classification

## Abstract

Worldwide, the cesarean section rate has steadily increased from 6.7% in 1990 to 21.1% in 2018 and is expected to rise even more. The World Health Organization propose the adoption of the Robson classification system as a global standard for monitoring, evaluating, and comparing delivery rates. The purpose of the current study is to use the Robson classification system to investigate how, independently of medical factors, the day of the week and time of delivery may be related to the mode of birth. In the sample analysis, we included the records of 8572 women giving birth in one private health facility in Greece. Over 60% of deliveries during the study period were performed by cesarean section, 30.6% by vaginal delivery, and 8.5% of deliveries were performed by operative vaginal delivery. The results of this study indicate that the lowest birth rates are observed on Monday, Saturday, and Sunday. Nulliparous women with no previous cesarean delivery, with a singleton in cephalic presentation ≥37 weeks with spontaneous labor (group 1) are 73% more likely to deliver by cesarean section between 08:00 A.M. and 03:59 P.M. compared to those who give birth between 12:00 A.M. and 07:59 A.M. Also, multiparous women with a single cephalic term pregnancy and one previous cesarean section (group 5.1) are 16.7 times more likely to deliver by cesarean section in the morning compared to overnight deliveries. These results point out two non-clinical variables that influences the CS rate. The Robson classification system was a useful tool for the above comparisons.

## 1. Introduction

Worldwide, strategies in childbirth aim to improve and ensure safety during perinatal care [1]. There is, however, growing evidence that labor has become a medical phenomenon without a clear cause in recent decades [2,3,4,5,6,7]. Globally, the cesarean section (CS) rate has steadily increased from 6.7% in 1990 to 21.1% in 2018 and is expected to rise to 28.5% by 2030 [8,9]. Cesarean section is a life-saving surgical procedure when performed appropriately and according to medical indication. However, high rates of cesarean births have become a controversial public health problem due to potential association with maternal and perinatal complications affecting the index or future pregnancies [4]. Such complications include postpartum hemorrhage, blood transfusion, urological complications, postpartum infections, stillbirths, morbidly adherent placenta, peripartum hysterectomy, and other direct and indirect perinatal complications [3,5,8,9]. In 2015, the World Health Organization (WHO) proposed the adoption of a universal CS classification system as a global standard for monitoring, evaluating, and comparing delivery rates within a health facility and/or among them [10]. The Robson classification divides pregnant women into one of 10 mutually exclusive groups based on five obstetric characteristics: parity, gestational age, onset of labor, number of fetuses, and fetal presentation [11]. It is worth noting that it is important to evaluate the underlying reasons for the increasing trend of CS in the light of the multidimensional nature of this phenomenon [7,12]. 

The decision to have a CS is influenced by both clinical and non-clinical factors [13]. Rates are closely linked to factors that affect women’s risk profile, medical indications, and non-medical reasons including economic, cultural, and social factors [14,15,16,17]. For instance, a non-clinical reason that could affect the decision on delivery type is the day and time of delivery. Both human and nonhuman primates share a maternal circadian mechanism that regulates birth patterns [18]. Findings of recent studies regarding the fundamental mechanisms of time of birth and regulation of the contractile machinery in the myometrium indicate that increased sensitivity to melatonin and oxytocin usually lead to human delivery [19,20]. There is limited information about the hourly distribution of birth in primate populations, including humans. Based on previous studies [19,20,21,22], there is a general consensus that women go into labor mainly at night without any medical interference. Due to the unpredictability of vaginal delivery, there is a higher likelihood of unwarranted CS, as it is the only mode of delivery in which the time can be planned [2,3].

Earlier studies have shown inconsistent results regarding non-clinical factors in relation to CS, highlighting the need for further investigation [23,24]. The purpose of the current study is to use the Robson classification system to investigate how, independently of medical factors, the day of the week and time of delivery may be related to the mode of delivery. Moreover, the present study aims to fill the gap that exists in the literature regarding the possible association between day of week and mode of delivery. The discovery of possible correlations may lead to the development of interventions aimed at reducing unnecessary cesarean sections caused by non-clinical factors.

## 2. Materials and Methods

The present study is part of a wider research protocol on the implementation of the Robson classification in the Greek classification setting. The methods of the research study have been previously published and are summarized here as follows [25]. This is a single-center retrospective study conducted between January and December 2019. The analysis is based on 7849 deliveries performed in a tertiary private hospital located in Athens, Greece. This hospital is a referral one; it handles all types of pregnancies (including high-risk pregnancies) from all regions of the country and it includes a neonatal intensive unit (NICU). Moreover, is estimated that it conducts approximately 10,000 deliveries every year. Specifically in 2019, 8681 deliveries were performed. Women with stillbirth fetuses (*n* = 73) were excluded from the sample.

Data collection was made from the medical records of women who had given birth from 22 weeks of gestation onward and with a neonate weighing at least 500 g. Data related to the day and time of delivery were retrieved from the electronic health record of women. Anonymity and de-identification were implemented before analysis. Written informed consent was not a prerequisite as all women sign a GDPR form at hospital admission. The study was approved by the Ethics Committee of the hospital (protocol code: 1146/24-09/20).

Data were expressed as mean ± standard deviation (SD) for continuous variables and as frequencies (*n*) and percentages (%) for categorical variables. The percentage of CS between hourly distributions of deliveries without intervention and with medical intervention (included CS) was compared using the chi-squared test. To adjust for potential confounders, a multifactorial logistic regression analysis was carried out to evaluate the contribution of the hourly birth pattern on cesarean delivery, including all potential confounders of clinical and demographic variables. The statistical analyses were conducted using IBM^®^SPSS^®^ software, version 22.0 (IBM Corporation). All analyses were two-tailed and a *p*-value < 0.05 was considered statistically significant.

## 3. Results

In our study, medical records and delivery data were collected from 8.572 women. In Table 1 is demonstrated maternal, newborn, and delivery characteristics. The majority of women were aged between 30 and 39 years old (71.0%), and were of Greek ethnicity (94.6%). A high percentage of the studied population (47.1%) gave birth between 37 + 0 and 38 + 6 weeks of gestation, and 39.3% between 39 + 0 and 41 + 6. Pregnancy was singleton in 95.6% of the sample and 57.6% of women were nulliparous. Among all deliveries, 60.9% of the sample had a cesarean section, 30.6% had a vaginal delivery, and 8.5% had an operative vaginal delivery (vacuum or/and forceps extraction). Regarding the newborns, 51.4% were males and most of them had a birth weight between 3000 and 3999 g (60.3%).

In Table 2 is demonstrated the Robson classification per mode of delivery. We categorized women into the 10 groups of the Robson classification. Every woman admitted to deliver was classified into one, and only one, of the 10 groups and no woman was left out of the classification. Women were classified according to the definition of obstetric variables (parity, previous CS, onset of labor, number of fetuses, gestational age, and fetal lie an presentation) and subdivisions for groups 2, 4, and 5 as proposed by WHO [11]. The four groups that accumulate the highest percentages amongst the total sample according to Robson classification are seen sequentially in group 2, group 5, group 4, and group 1. More specifically, groups 2a + 2b comprise 34.5% (2956/8572) of the sample, groups 5.1 + 5.2 comprise 19.8% (1702/8572) of the sample, groups 4a + 4b account for 12.0% (1036/8572) and group 1 accounts for 10.8% of the study population (928/8572). In group 1 of the Robson classification (nulliparous women with a single cephalic pregnancy at ≥37 weeks of gestation in spontaneous labor), 42% of the women gave birth by vaginal delivery, 38.7% by cesarean section, and 19.3% by operative vaginal delivery. Considering group 2a of the Robson classification (nulliparous women with a single cephalic pregnancy at ≥37 weeks of gestation who had labor induced) we observe a vaginal delivery in 46.4% of the women, a cesarean section in 32.7% of the women, and an operative vaginal delivery in 20.9% of the women. However, a higher rate for group 2 is noted for nulliparous women with a single cephalic pregnancy at ≥37 weeks of gestation who were admitted and delivered by pre-labor CS (group 2B) with a rate of 14.5%. For the multiparous women without previous CS, the majority (904 women) had labor induced (using any method, such as misoprostol, oxytocin, amniotomy, or foley endocervical catheter or other) and proceeded to vaginal delivery (group 4). As for group 5.1 (all multiparous women with one previous CS, with a single cephalic pregnancy, ≥37 weeks of gestation), the vast majority of women (93.3%) gave birth by a cesarean section whereas only 6.7% delivered vaginally after cesarean section (VBAC). In addition, particularly high CS rates were observed for nulliparous and multiparous women with single breech pregnancies (group 6 and 7) with rates of 99.% and 96.2%, respectively. Women with multiple gestation that fall under group 8 gave birth by vaginal delivery at a rate of 1.8% and by operative vaginal delivery at a rate of 1.3%, while the majority of these women gave birth by a cesarean section at a rate of 96.8%. For women with a single pregnancy with a transverse or oblique lie (including women with previous CS) the percentage of CS rate was 100%. Lastly, in group 10 which includes all women with a single cephalic pregnancy at <37 weeks of gestation (including women with previous CSs) the percentage of cesarean sections goes as high as 79.1%.

Furthermore, it was detected that 60.5% of all births occur from Monday to Thursday. Higher percentages were observed on Friday (1585/7849) and Tuesday (1340/7849) while lower percentages were noted on Saturday (1046/7849), Sunday (471/7849) and Monday (1035/7849). Totals of 1128 and 1244 women out of 7849 gave birth on Wednesday and Thursday, respectively (Figure 1). In addition, the distribution of the day by Robson classification group shows that in all groups (with the exception of group 3) Sunday is the day with the lowest birth rates (Table 3). Finally, 79.3% of all modes of deliveries took place during the daytime (08:00 A.M.–07:55 P.M.) with peaks in the morning/daylight hours and decreasing rates in the evening and night (Figure 2).

In this study, we examined additionally whether the mode of delivery between two types, vaginal delivery (including operative vaginal delivery and VBAC) and cesarean section, is related to time periods (Table 4). Indeed, a significant difference in the percentage of cesarean sections was observed among the explored time periods (*p* < 0.005). Comparisons by pairs show the divergence between time periods: 08:00 A.M.–03:59 P.M. compared to 12:00 A.M.–07:59 A.M. (*p* < 0.005, *p*_bonferroni_ < 0.005) and 04:00 P.M.–11:59 P.M. (*p* < 0.005, *p*_bonferroni_ < 0.005). Also, between 12:00 A.M.–07:59 A.M. and 04:00 P.M.–11:59 P.M. (*p* < 0.005, *p*_bonferroni_ < 0.005).

The current analysis revealed a significant difference between time period and Robson classification (Table 5 and Table 6). More specifically, the current analysis revealed a significant difference between time period 12:00 A.M.–07:59 A.M. and 08:00 A.M.–03:59 P.M. [adjusted OR (95% CI): 1.76 (1.22–2.54), *p* < 0.005] and between time period 12:00 A.M.–07:59 A.M. and 04:00 P.M.–11:59 P.M. [adjusted OR (95% CI): 2.35 (1.57–3.53), *p* < 0.005] of likelihood of cesarean delivery for Robson group 1. By applying logistic regression analysis models for Robson 1, it becomes apparent that labors performed in the period 08:00 A.M.–03:59 P.M. are 73% more likely to be carried out through a cesarean section compared to those performed between 12:00 A.M. and 07:59 A.M., and labors performed within the time period 04:00 P.M.–11:59 P.M. are 2.2 times more likely to be carried out through a cesarean section compared to those performed between 12:00 A.M. and 07:59 A.M.

Similarly for group 2a, labors performed between 08:00 A.M. and 03:59 P.M. are 2.2 times more likely to be carried out via a cesarean section compared to those performed in the interval 12:00 A.M.–07:59 A.M. whereas labors performed between 04:00 P.M. and 11:59 P.M. are 3 times more likely to be carried out via cesarean section compared to those performed in the interval 12:00 A.M.–07:59 A.M.

Regarding Robson 5.1, a substantial finding was that labors performed between 08:00 A.M. and 03:59 P.M. were 16.7 times more likely to be carried out through cesarean section compared to those performed in the interval 12:00 A.M.–07:59 A.M. Lastly, in Robson 10, labors performed in the time period 08:00 A.M.–03:59 P.M. were four times more likely to be carried out through cesarean section compared to those performed in the interval 12:00 A.M.–07:59 A.M.

## 4. Discussion

In this retrospective study we demonstrated that over 60% of deliveries during the study period were performed by a cesarean section, 30.6 by vaginal delivery, and 8.5% of deliveries were performed by operative vaginal delivery. It was observed that groups 2 and 5 of the Robson classification were linked to higher cesarean section rates. Based on international literature and WHO recommendations, these percentages are particularly high [8].

Our earlier study showed that cephalopelvic disproportion and previous cesarean section are the most frequent indications for a cesarean section [25]. Several studies have associated high cesarean section rates with non-clinical factors. For example, the type of hospital [26], the age and gender of the doctor [23], and economic factors [27] have been correlated with an increased probability of performing CS. In the present study, we sought to examine non-clinical factors related to high cesarean section rates, such as the day and time of delivery. The impact of the day on the delivery mode has not been extensively explored in previous studies. The results of this study indicate that the lowest birth rates are observed on Monday, Saturday, and Sunday. The analysis through the Robson classification allowed us to understand the distribution of births by day for each group separately, identifying women that disproportionately contribute to the high CS rate. Furthermore, monitoring specific groups will enable interventions that may lead to a reduction in unnecessary CS. The low rates of Sunday deliveries for all Robson groups in our study sample could be attributed to the particularly high rates of “scheduled” deliveries. For instance, groups 5.1 to 9 show cesarean rates more than 90%. Moreover, group 1 appears to have an equal distribution of births during the week. This is probably due to the fact that women in this category were in spontaneous labor. However, an interesting finding was that, out of the nulliparous women with a single cephalic pregnancy at ≥37 weeks of gestation, only 928/3884 were in spontaneous labor (Robson 1), while 1709/3884 and 1247/3884 had induced labor (Robson 2a) and planned cesarean section (Robson 2b), respectively. Likewise, a high rate of labor induction was also observed in multiparous women without a previous scar, with a single cephalic pregnancy, at >37 weeks of gestation (1007/1408), with the higher rates being noted from Tuesday to Saturday. Similarly, category 10 of the Robson classification, which includes all women with a single, cephalic pregnancy at <37 weeks of gestation, including women with previous scars, presents a lower number of deliveries on Sunday which reveals the “scheduled” nature of preterm births rather than the state of emergency it should bear.

To our knowledge, this is the first study to present the daily distribution of births by Robson classification. Therefore, there is no previous evidence to compare our data with. In an earlier study conducted in three hospitals (two public and one private) in Greece, it seemed that, in the private hospital, there was a rapid decrease in cesarean sections on Sunday while, in the two public hospitals, there was a decrease in cesarean sections and an increase in vaginal deliveries on the weekdays [28]. A recent study in America showed that women who give birth during weekends are 27% less likely to have CS, compared to those who give birth on weekdays [29]. Furthermore, in our study, the hourly distribution of labor was not stable and shows a de-escalating trend during the 24 h. Between 12:00 A.M. and 07:59 A.M., only 1101/7849 deliveries were performed, with vaginal delivery rates of 58.5% and cesarean section rates of 41.5%. Our results are not in agreement with previously published data from England [30] and Austria [31], showing overnight delivery rates up to 55.8% and 49.2%, respectively. Of the total number of vaginal deliveries (*n* = 2625), the minority of them (*n* = 644) were performed between 12:00 A.M. and 07:59 P.M. while the majority of them (*n* = 1328 women) were performed between 08:00 A.M. and 03:59 P.M. In addition, our findings are in contrast with the results of previous studies reporting higher rates of vaginal delivery during the night [30,31,32].

Furthermore, our findings do not seem to follow the biological expression of childbirth which is related to the increase in the secretion of melatonin and oxytocin during the night, resulting in higher rates of vaginal delivery during night hours [33,34]. In a recent study by Kanwar et al., it is summarized that night deliveries are not only numerically more common but appear to be also physiologically more efficient than day-onset deliveries [32]. In an earlier study conducted in Greece, it was shown that the time of delivery is a significant factor in the increase in cesarean sections with >70% of them being performed between 8:00 A.M. and 4:00 P.M. [28]. The comparison between time of delivery and mode of delivery raises questions about which groups of women influence this association. The Robson classification gave us a deeper interpretation of our results. Our results, unfortunately, highlight the medicalization of childbirth as women in Robson category 1 are 73% more likely to deliver by cesarean section between 08:00 A.M. and 3:59 P.M. compared to those who give birth between 12:00 A.M. and 07:59 A.M. Lastly, women in group 5.1 are 16.7 times more likely to deliver by cesarean section in the morning compared to overnight deliveries. This is due to the particularly high rates of repeat cesarean section and the low rates of VBAC in the Greek population as has also been supported by previous studies [28,35]. Our results align with this finding, indicating that in Greece healthcare policies and obstetrics practices need more effective strategies in labor management. Moreover, further qualitative research which studies obstetrician practices and beliefs, and financial incentives would be useful in elucidating the relationship between obstetricians’ convenience and CS rates.

### Strengths and Limitations

Our work presents some strengths and limitations. First, the sample size of our study was large enough to achieve an annual representative sample of births in Greece that is considered representative of the Greek population; the hospital that approved this study is the largest private obstetrical clinic in Greece and, consequently, serves as a referral hospital having a full complement of services, including obstetrics, neonatology, and intensive care units, strengthening, therefore, the representativeness of our sample. Second, it was the first time that non-clinical factors were studied by applying the Robson classification in Greece. We are aware, though, that our research had some limitations as well. The main limitations of our study were its retrospective nature and the fact that it was conducted in a single hospital. Additionally, we could not account for all non-clinical confounders. Unfortunately, we were unable to study the level of urgency for the performed cesarean sections because the available data were of heterogeneous quality which could lead us to misleading conclusions.

## 5. Conclusions

To conclude, the findings of the present study add significant information on the association of non-clinical variables with the increasing trend of cesarean section in the Greek setting. Robson’s classification appears to be an effective tool for studying, in a standardized manner, non-clinical variables associated with the mode of delivery. Further investigation of additional non-clinical variables through qualitative research would be a useful tool. A better understanding of the multifactorial nature of the increased CS rate will contribute to quality improvement in hospitals to ensure the provision of equal and impartial perinatal care.

## Figures and Tables

**Figure 1 healthcare-11-02158-f001:**
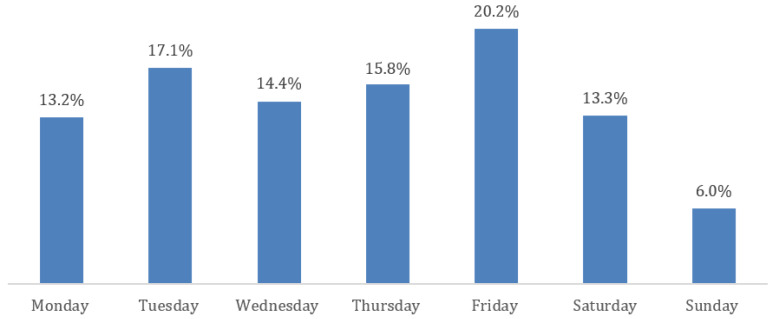
Day distribution of births.

**Figure 2 healthcare-11-02158-f002:**
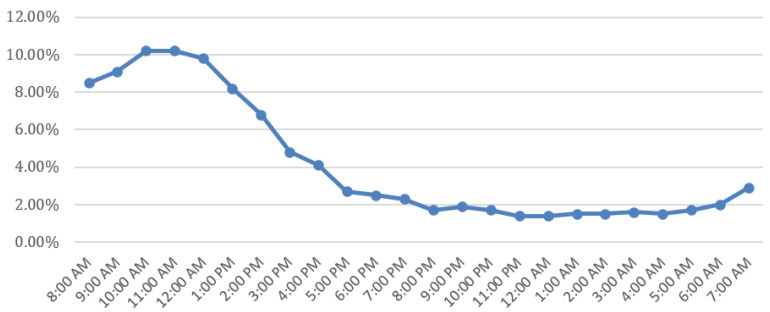
Hourly distribution of births.

**Table 1 healthcare-11-02158-t001:** Maternal, newborn, and labor characteristics (N = 8681).

Characteristics		Frequency (*n*)	Percentage (%)
Maternal age (years)	mean ± SD (Min, Max)	34.16 ± 4.90 (17–52)	
	<20	16	0.2
	20–29	1392	16.2
	30–39	6084	71.0
	≥40	1080	12.6
Nationality	Other	464	5.4
	Greek	8108	94.6
Gestational age (weeks)	<37^+0^	1160	13.5
	37^+0^–38^+6^	4039	47.1
	39^+0^–41^+6^	3367	39.3
	≥42^+0^	6	0.1
Type of pregnancy	Single	8194	95.6
	Multiple	378	4.4
Parity	0	4938	57.6
	1	3030	35.3
	≥2	604	7.0
Mode of delivery	Vaginal delivery	2620	30.6
	Operative vaginal delivery	728	8.5
	Cesarean section	5224	60.9
Newborn’s sex	Male	4596	51.4
	Female	4351	48.6
Newborn’s birth weight (g)	<2500	1055	11.8
	2500–2999	2234	25.0
	3000–3999	5397	60.3
	≥4000	261	2.9

**Table 2 healthcare-11-02158-t002:** Distribution of vaginal deliveries, operative vaginal deliveries, and cesarean sections in each Robson classification.

		Mode of Delivery							
		Vaginal Delivery		Operative Vaginal Delivery		Cesarean Section		Total	
		Count	% within Robson	Count	% within Robson	Count	% within Robson	Count	% within Robson
Robson Class	1	390	42.0%	179	19.3%	359	38.7%	928	10.8
	2a	793	46.4%	358	20.9%	558	32.7%	1709	20.0
	2b	0	0.0%	0	0.0%	1247	100.0%	1247	14.5
	3	330	88.7%	30	8.1%	12	3.2%	372	4.3
	4a	904	89.8%	83	8.2%	20	2.0%	1007	11.7
	4b	0	0.0%	0	0.0%	29	100.0%	29	0.3
	5.1	60	4.0%	40	2.7%	1387	93.3%	1487	17.3
	5.2	2	0.9%	0	0.0%	213	99.1%	215	2.5
	6	1	0.4%	1	0.4%	278	99.3%	280	3.3
	7	1	0.9%	3	2.8%	102	96.2%	106	1.2
	8	7	1.8%	5	1.3%	367	96.8%	379	4.4
	9	0	0.0%	0	0.0%	43	100.0%	43	0.5
	10	132	17.1%	29	3.8%	609	79.1%	770	9.0

**Table 3 healthcare-11-02158-t003:** Distribution of day in each Robson classification.

Robson Classification													
Day Distribution	1	2a	2b	3	4a	4b	5.1	5.2	6	7	8	9	10
	Ν (%)	Ν (%)	Ν (%)	Ν (%)	Ν (%)	Ν (%)	Ν (%)	Ν (%)	Ν (%)	Ν (%)	Ν (%)	Ν (%)	Ν (%)
Monday	101 (13.5)	176 (13.0)	157 (12.6)	50 (14.6)	83 (9.0)	4 (13.8)	255 (15.5)	33 (15.3)	34 (12.2)	13 (12.6)	52 (13.9)	4 (9.3)	103 (13.9)
Tuesday	103 (13.8)	237 (17.5)	254 (20.4)	59 (17.2)	148 (16.0)	6 (20.7)	248 (17.1)	40 (18.6)	38 (13.6)	15 (14.6)	73 (19.5)	7 (16.3)	112 (15.1)
Wednesday	115 (15.4)	209 (15.5)	164 (13.2)	53 (15.5)	149 (16.1)	6 (20.7)	204 (14.1)	28 (13.0)	38 (13.6)	10 (9.7)	44 (11.8)	4 (9.3)	104 (14.0)
Thursday	97 (13.0)	183 (13.5)	216 (17.3)	32 (9.3)	139 (15.0)	4 (13.8)	247 (17.1)	44 (20.5)	51 (18.3)	18 (17.5)	66 (17.6)	13 (30.2)	134 (18.1)
Friday	131 (17.5)	293 (21.7)	264 (21.2)	64 (18.2)	199 (21.5)	5 (17.2)	309 (21.4)	40 (18.6)	61 (21.9)	29 (28.2)	69 (18.4)	6 (14.0)	115 (15.5)
Saturday	111 (14.8)	173 (12.8)	171 (13.7)	42 (12.2)	138 (14.9)	4 (13.8)	165(11.4)	20 (9.3)	40 (14.3)	15 (14.6)	47 (12.6)	8 (16.8)	112 (15.1)
Sunday	91 (12.1)	81 (6.0)	21 (1.7)	43 (12.5)	71 (7.7)	0 (0)	49(3.4)	10 (4.7)	17 (6.1)	3 (2.9)	23 (6.1)	1 (2.3)	61 (8.2)

**Table 4 healthcare-11-02158-t004:** Distribution of time between vaginal delivery and cesarean section.

			Mode of Delivery		Total
			Vaginal Delivery	Cesarean Section	
Time period	12:00 A.M.–07:59 A.M.	Count	644	457	1101
		% within ΤΙΜΕ	58.5%	41.5%	100.0%
	08:00 A.M.–03:59 P.M.	Count	1328	3983	5311
		% within ΤΙΜΕ	25.0%	75.0%	100.0%
	04:00 P.M.–11:59 P.M.	Count	653	784	1437
		% within ΤΙΜΕ	45.4%	54.6%	100.0%
Total		Count	2625	5224	7849
		% within ΤΙΜΕ	33.4%	66.6%	100.0%

**Table 5 healthcare-11-02158-t005:** Time of delivery compared to mode of delivery per Robson group.

Robson Class			Delivery				*p*-Value
			Vaginal Delivery		Cesarean Section		
			N	%	N	%	
1	Τime period	12:00 A.M.–07:59 A.M.	128	63.7%	73	36.3%	<0.005
		08:00 A.M.–03:59 P.M.	170	50.3%	168	49.7%	
		04:00 P.M.–11:59 P.M.	92	43.8%	118	56.2%	
2a	Τime period	12:00 A.M.–07:59 A.M.	142	75.9%	45	24.1%	<0.005
		08:00 A.M.–03:59 P.M.	410	59.2%	283	40.8%	
		04:00 P.M.–11:59 P.M.	242	51.3%	230	48.7%	
2b	Τime period	12:00 A.M.–07:59 A.M.	-	-	54	100.0%	---
		08:00 A.M.–03:59 P.M.	-	-	1126	100.0%	
		04:00 P.M.–11:59 P.M.	-	-	67	100.0%	
3	Τime period	12:00 A.M.–07:59 A.M.	133	97.8%	3	2.2%	---
		08:00 A.M.–03:59 P.M.	133	96.4%	5	3.6%	
		04:00 P.M.–11:59 P.M.	65	94.2%	4	5.8%	
4a	Τime period	12:00 A.M.–07:59 A.M.	176	98.3%	3	1.7%	---
		08:00 A.M.–03:59 P.M.	546	98.7%	7	1.3%	
		04:00 P.M.–11:59 P.M.	185	94.9%	10	5.1%	
4b	Τime period	08:00 A.M.–03:59 P.M.	-	-	26	100.0%	---
		04:00 P.M.–11:59 P.M.	-	-	3	100.0%	
5.1	Τime period	12:00 A.M.–07:59 A.M.	21	20.0%	84	80.0%	<0.005
		08:00 A.M.–03:59 P.M.	18	1.5%	1201	98.5%	
		04:00 P.M.–11:59 P.M.	21	17.1%	102	82.9%	
5.2	Τime period	12:00 A.M.–07:59 A.M.	2	13.3%	13	86.7%	---
		08:00 A.M.–03:59 P.M.	0	0.0%	184	100.0%	
		04:00 P.M.–11:59 P.M.	0	0.0%	16	100.0%	
6	Τime period	12:00 A.M.–07:59 A.M.	1	2.4%	40	97.6%	---
		08:00 A.M.–03:59 P.M.	0	0.0%	207	100.0%	
		04:00 P.M.–11:59 P.M.	0	0.0%	31	100.0%	
7	Τime period	12:00 A.M.–07:59 A.M.	1	11.1%	8	88.9%	---
		08:00 A.M.–03:59 P.M.	0	0.0%	77	100.0%	
		04:00 P.M.–11:59 P.M.	0	0.0%	17	100.0%	
8	Τime period	12:00 A.M.–07:59 A.M.	1	2.0%	48	98.0%	---
		08:00 A.M.–03:59 P.M.	3	1.1%	260	98.9%	
		04:00 P.M.–11:59 P.M.	3	4.8%	59	95.2%	
9	Τime period	12:00 A.M.–07:59 A.M.	-	-	4	100.0%	---
		08:00 A.M.–03:59 P.M.	-	-	36	100.0%	
		04:00 P.M.–11:59 P.M.	-	-	3	100.0%	
10	Τime period	12:00 A.M.–07:59 A.M.	39	32.2%	82	67.8%	<0.005
		08:00 A.M.–03:59 P.M.	48	10.6%	403	89.4%	
		04:00 P.M.–11:59 P.M.	45	26.6%	124	73.4%	

**Table 6 healthcare-11-02158-t006:** Logistic regression of the indicator delivery (vaginal delivery vs. cesarean section) compared to time of delivery (*).

		Crude OR	95% CI		*p*-Value	Adjusted OR	95% CI		*p*-Value
	Τime Period								
Total					<0.005				<0.005
	12:00 A.M.–07:59 A.M.	1.00				1.00			
	08:00 A.M.–03:59 P.M.	4.23	3.69	4.84	<0.005	4.53	3.93	5.24	<0.005
	04:00 P.M.–11:59 P.M.	1.69	1.44	1.98	<0.005	1.98	1.67	2.34	<0.005
Robson 1					<0.005				<0.005
	12:00 A.M.–07:59 A.M.	1.00				1.00			
	08:00 A.M.–03:59 P.M.	1.73	1.21	2.47	<0.005	1.76	1.22	2.54	<0.005
	04:00 P.M.–11:59 P.M.	2.25	1.51	3.34	<0.005	2.35	1.57	3.53	<0.005
Robson 2a					<0.005				<0.005
	12:00 A.M.–07:59 A.M.	1.00				1.00			
	08:00 A.M.–03:59 P.M.	2.18	1.51	3.14	<0.005	2.10	1.44	1.86	<0.005
	04:00 P.M.–11:59 P.M.	3.00	2.05	4.38	<0.005	2.79	1.89	4.12	<0.005
Robson 5.1					<0.005				<0.005
	12:00 A.M.–07:59 A.M.	1.00				1.00			
	08:00 A.M.–03:59 P.M.	16.68	8.56	32.50	<0.005	19.17	8.98	40.90	<0.005
	04:00 P.M.–11:59 P.M.	1.21	0.62	2.37	0.570	1.85	0.83	4.12	0.133
Robson 10					<0.005				<0.005
	12:00 A.M.–07:59 A.M.	1.00				1.00			
	08:00 A.M.–03:59 P.M.	4.00	2.46	6.48	<0.005	4.12	2.51	6.78	<0.0005
	04:00 P.M.–11:59 P.M.	1.31	0.79	2.18	0.300	1.54	0.91	2.62	0.111

(*) Adjusted for: maternal age, gestational age, smoking, newborn’s birth weight.

## Data Availability

Not applicable.

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
