# Peer review of "Can the Day of the Week and the Time of Birth Predict the Mode of Delivery According to Robson Classification?"

_healthcare, 2023, doi:10.3390/healthcare11152158_

Round 1

Reviewer 1 Report

I think it is a very interesting study, very well structured and very well done. It is important to carry out this type of study in order to reduce those caesarean sections that can be avoided, as the recovery of women will be faster with less comorbidity.

Reviewer 2 Report

Thank you for the opportunity to write a review.

Original work, very interesting, well presented results.

Properly drawn conclusions from research. I consider the selection of the group to be correct, the number of women surveyed deserves attention.

I have no comments. Congratulations on your valuable research

Reviewer 3 Report

Thank you for inviting me to review manuscript 2495286 under consideration at Healthcare. This study uses the Robson classification system to investigate how, independently of medical factors, the day of the week and time of delivery may affect the mode of birth. Using records of 8572 women giving birth in one private health facility in Greece, they find that , the day of the week and time of delivery may affect the mode of birth.

Introduction

·       In the introduction, I would recommend that the authors remove the first two sentences (lines 27-29) and start with “Worldwide…” which makes more sense for the framing of the issue.

·       I would also encourage the authors to address why this research is important. In other words, why is it important and relevant to understand mode of birth and why we should we be concerned about rising levels of CSs?

·       The authors mention “cultural factors may have influenced natural adaptation”. Can you explain what you mean and why this is relevant to discuss in the introduction? I got really lose in lines 46-49.

Materials and Methods

·       Although the authors explain that the present study is part of a wider research protocol on the implementation of the Robson classification in the Greek setting the authors do not explain how mode of birth and birth outcomes at a private hospital in Athens, Greece, might be different than a public hospital or a private hospital in a secondary city. For example, more CS in private hospitals because they might attract more affluent women, more insured women, etc. Can you provide more information and/or include this as a potential limitation later in the paper?

Results

·       The authors present the breakdown of mode of birth in this section. While they do so in the abstract too, they do not include “30,6% had a vaginal delivery” in the abstract, which I think would be useful to include alongside the incidence of CS and operative vaginal delivery. Same for the first line of the discussion.

·       In table 1, I was very confused by the line “34,16±4,90 ( 17-52)”. Is this a typo?

·       Line 95 needs editing “In Table 2 is demonstrated…”

·       The authors gloss over how they classified women into the Robson classification and the results in Table 2. More information should be provided on this.

·       The author mention VBAC in line 134 for the first time, without previous mention. What does this mean?

Discussion

·       The manuscript needs a stronger discussion overall.

·       It is very unclear to me the Robson component to this study. What does this add to our understanding of mode of birth by look at day and time?

·       The authors discuss the importance of this study in filling gaps in the literature (177-178), but this should be included at the beginning (introduction) to justify this study.

·       Why are hospitals and doctors pushing for CS deliveries?

English language quality was sufficiently good, but some minor edits will be required

Reviewer 4 Report

Too bad that the article is a retrospective study in a single center, even if this does not affect the quality of the analysis. Curious that the CS rate is 60%, out of any world standard. I would have expected 25%-30% in a big city. However, the article is well written and the data are interesting for understanding how people give birth in Athens, the capital of the Greek state.

The English language is fine, only text editing needs to be checked, as there are some spacing and typo errors.

Reviewer 5 Report

In the abstract, a simple explanation of the classification should be given, such as category 1, group 5.1.

Figure 2 can be expressed in 24 hours without the additional text.

"The purpose of the current study is to use the Robson classification system to investigate how, independently of medical factors, the day of the week and time of delivery may affect the mode of delivery." Is the date and time of delivery a factor in the mode of delivery or a manifestation of the outcome? For example, delivery is a long process, and the timing of the final delivery is only one of the milestones in this process. Or, it can be expressed here as the relationship between them.

"The methods of the research study have been previously published and are summarized here as follows." Here, references need to be added.

Some commas need to be corrected, e.g. 71,0% (71.0%).

在摘要中,应给出分类的简单解释,例如第1类,第5.1组。

Figure 2 can be expressed in 24 hours without the additional text.

"The purpose of the current study is to use the Robson classification system to investigate how, independently of medical factors, the day of the week and time of delivery may affect the mode of delivery.“
分娩日期和时间是分娩方式的一个因素还是结果的一种表现?例如,交付是一个漫长的过程,最终交付的时间只是这个过程中的里程碑之一。或者,它可以在这里表示为它们之间的关系。

"The methods of the research study have been previously published and are summarized here as follows.“的
在此,需要添加参考文献。

71.0%(71.0%)。 71,0% (71.0%).
